# ON THE ROBUSTNESS OF SCRNA-SEQ FOUNDATION MODELS FOR PLANTS UNDER CROSS-DOMAIN EXPERIMENTAL SHIFT

**Manuel Fernández Burda**[1,2]        **Rodrigo Bonazzola**[1,3]        **Georgina Stegmayer**[1]

**Enzo Ferrante**[2,4]        **Diego H. Milone**[1]

[1]Research Institute for Signals, Systems and Computational Intelligence, sinc(i), FICH-UNL, CONICET. [2]Laboratory of Applied Artificial Intelligence (LIAA), Institute of Computer Sciences (ICC), CONICET - UBA    [3]EMBL-EBI    [4]Apolo Biotech

## ABSTRACT

Foundation models for single-cell transcriptomics promise to learn generalizable representations of cellular states, yet their robustness to cross-study distribution shift remains underexplored in plant systems. We introduce scAraFM, an Arabidopsis-specific foundation model, and evaluate its utility for stress prediction across leaf and root scRNA-seq datasets under three increasingly challenging protocols: random splits from a single experiment, replicate-based splits from a single experiment, and cross-experiment transfer learning. For single experiment settings, we find that random splits can overestimate performance by 20–30 AUROC points compared to replicate-held-out evaluation and across independent experiments, underscoring the need for study-aware validation in fragmented transcriptomic landscapes. Across representation strategies, gene-identity-preserving features consistently outperform pooled summaries, even when the latter are derived from pretrained transformers. Notably, simple baselines using raw reads remain competitive or superior to learned embeddings under single-experiment scenarios, challenging claims of universal advantage for foundation-model-derived features. Our promising results on cross-experiment transfer learning emphasize that evaluation design is as critical as model architecture, and that preserving per-gene structure aids generalization in downstream tasks.

## 1    INTRODUCTION

Single-cell RNA sequencing (scRNA-seq) is transforming molecular biology by resolving tissue composition and stress responses at cellular resolution. Yet plant scRNA-seq datasets remain fragmented across laboratories, protocols, and study goals, making robust generalization a central challenge. In practice, many downstream classifiers are still assessed under random train/test splits, which can overestimate performance by allowing study-specific signatures (batch effects, annotation conventions, and lab-specific processing) to leak across partitions. Best-practice guidelines in scRNA-seq analysis emphasize that such domain-specific effects can dominate apparent structure unless explicitly controlled for (Luecken & Theis, 2019; Haghverdi et al., 2018).

Foundation-model-style representation learning has recently been adapted to transcriptomic data by treating genes as tokens and cells as sequences, enabling self-supervised pretraining followed by task-specific adaptation (Bommasani et al., 2021). Prominent examples include scBERT (Yang et al., 2022), scGPT (Cui et al., 2024), and Geneformer (Theodoris et al., 2023). While this paradigm has been developed primarily for human and mouse single-cell atlases, plant-focused counterparts are comparatively scarce and less systematically evaluated, particularly for robustness under cross-study shifts (Chau et al., 2024; Cao et al., 2025). For plant biology, stress classification provides a particularly pragmatic robustness test: the same nominal stress label can be realized under different genotypes, protocols, time points, and experimental designs, so models must generalize beyond a

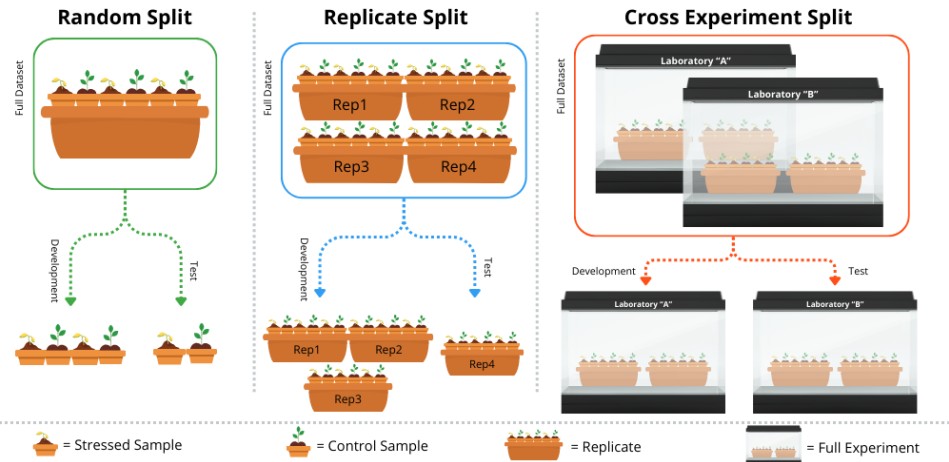

Figure 1: **Schematic of evaluation protocols for stress classification using foundational models.** We compare three dataset-splitting strategies with increasing resistance to study-specific confounders. Icons denote stressed vs. control samples, replicates, and full experiments.

single study signature. Recent re-evaluations highlight that simple linear models on raw per-gene reads can be surprisingly strong: a logistic regression baseline matches or outperforms scBERT on cell-type annotation across datasets, including few-shot settings (Boiarsky et al., 2024). This motivates the use of well-tuned classical baselines as rigorous reference points when assessing the added value of learned representations.

Here, we develop and evaluate an *Arabidopsis thaliana* foundation model (scAraFM) as a feature extractor for stress prediction in leaf and root scRNA-seq, and build an evaluation framework that isolates the effect of distribution shift under the same downstream task. Concretely, we (i) benchmark three increasingly challenging protocols: random, replicate-held-out, and cross-experiment transfer learning (Fig. 1); (ii) compare multiple cell-level feature constructions from per-gene embeddings; and (iii) test whether learned representations improve robustness relative to strong transcriptomic baselines (raw reads of highly variable genes). Our findings caution that in-distribution scores can be misleadingly optimistic, and suggest that preserving gene identity in representations is a key ingredient for transferable signal.

## 2  MATERIALS & METHODS

**Data.** We assembled a compendium of publicly available *Arabidopsis thaliana* transcriptomic datasets from major community repositories (scPlantDB (He et al., 2024) and NCBI GEO (Edgar et al., 2002)), prioritizing single-cell RNA-seq studies with well-annotated experimental metadata to enable cross-study evaluation. To support downstream stress-related benchmarking, we additionally curated a focused subset of datasets (Supplementary Table 1) covering diverse perturbations (drought, pathogen infection, osmotic stress, heat shock, sugar response, gravitropic reorientation), stratified by tissue (leaf vs. root) and organized into the evaluation protocols of our experimental design (described in the following section). All scRNA-seq datasets were processed with a standardized Scanpy-based (Wolf et al., 2018) pipeline to ensure a consistent feature space across studies and to make cross-experiment comparisons well-posed (refer to Appendix A.1.1 for a detailed explanation of the pre-processing).

**Foundation model & representation strategies.**  scAraFM inherits its architecture from scBERT (Yang et al., 2022), where each gene is represented by a learned embedding initialized from Gene2Vec, and expression values modulate these embeddings before they are fed to the transformer encoder. We carefully curate, harmonize, and perform quality control of a large-scale *Arabidopsis*-specific pretraining corpus (Supplementary Table 2), comprising ~834k leaf and ~878k root cells across 26 public datasets. By pretraining on Arabidopsis-specific transcriptomes, scAraFM learns gene co-expression patterns and regulatory structure native to the target organism, which we hypoth-

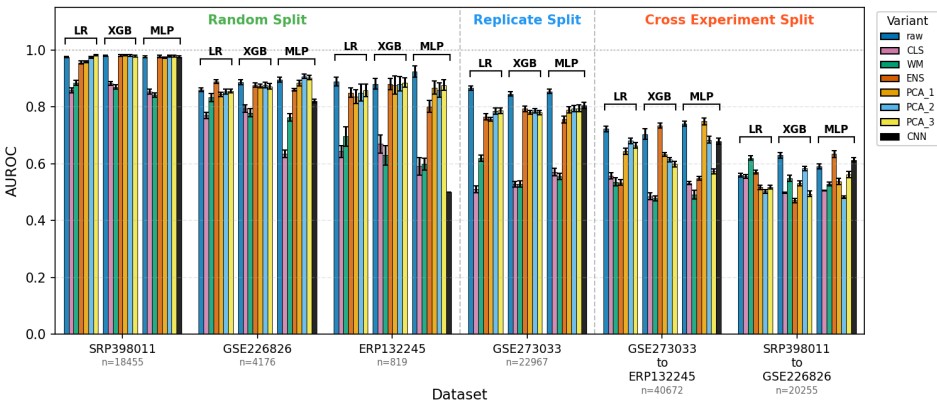

Figure 2: **Stress classification performance on varying scenarios for *Arabidopsis* leaves datasets.** Each bar depicts AUROC for each of the classifiers implemented (LR, XGBoost, and MLP) along raw counts and varying embedding combination methodologies. On the columns, the classification tasks are shown in rising difficulty: random splits, replicate splits and cross-experiment splits. Both datasets in each cross-experiment design share the same stressing conditions. Error bars are shown from bootstrapped test samples.

esize improves downstream transfer for plant stress classification. We extract cell-level features from scAraFM embeddings by aggregating gene-level features using three complementary approaches:

**(1) CLS token:** The special classification token embedding $\mathbf{h}_0 \in \mathbb{R}^d$ from the final layer, where $d = 200$ is the model dimension.

**(2) Expression-weighted mean (WM):** A weighted average of gene embeddings, normalized by total expression. For each cell $c$, $\mathrm{WM}_c = \frac{1}{Z} \sum_{g=1}^{n_\mathrm{g}} \mathbf{h}_g x_g$, where $x_g$ is the normalized expression of gene $g$ and $Z = \sum_g x_g$.

**(3) Per-gene embeddings:** The full concatenated representation $\mathrm{H}_c = [\mathbf{h}_1 \,|\, ... \,|\, \mathbf{h}_{n_\mathrm{genes}}] \in \mathbb{R}^{n_\mathrm{genes} \times d}$, preserving gene identity. To condense the high dimensionality input for classifiers ($n_\mathrm{genes} \times d$), we test three aggregation strategies. **(3a) Ensemble (ENS):** Train $d$ separate classifiers, each on a single embedding dimension across all genes, then combine via majority voting. **(3b) PCA dimensionality reduction:** Reduce embedding dimensions to their top $k$ principal components ($k \in \{1, 2, 3\}$ in our experiments and PCA is fitted on the training set), yielding features of size $n_\mathrm{genes} \times k$. **(3c) CNN:** Finetune a convolutional head that processes gene embeddings via 1D convolution (pooling down the embedding dimension) followed by a three-layer MLP, as in scBERT (Yang et al., 2022).

For all representations except (3c), we train three different classifier heads: logistic regression (LR), a multi-layer perceptron (MLP) (Pedregosa et al., 2011) and XGBoost (Chen & Guestrin, 2016).

**Experimental design.** We benchmark the models on a stress-classification task under three increasingly stringent evaluation protocols (Fig. 1): (i) *random split*, where train/test are sampled from a single experiment; (ii) *replicate-based split*, where one replicate is held out for testing and the remaining replicates are used for training; and (iii) *cross-experiment split*, where training and testing are performed on two independent datasets measuring the same stress condition, thereby maximizing domain shift and probing cross-study generalization. To probe performance across data regimes, we train with capped training-set sizes of 1,000, 5,000, 10,000, and 20,000 cells (when possible) using a fixed, class-balanced test set, and additionally train once using all available training data after holding out the test set; full protocol details are provided in Appendix A.2.

## 3 RESULTS

### 3.1 LEAF TISSUE

Our setup reveals a clear difficulty gradient (Figure 2): AUROC is highest under random splits, drops when a replicate is held out, and declines most sharply under cross-experiment transfer. This ordered degradation confirms that study-specific confounders inflate within-dataset scores and that cross-experiment evaluation provides the most stringent assessment of generalization.

Additionally, across all experimental designs we observe a consistent pattern: models that preserve gene identity in their input representation perform better than those that compress gene identities into a single pooled vector. In particular, the condensed schemes WM and CLS underperform, whereas approaches that retain per-gene structure (ENS, PCA, and CNN) achieve consistently stronger results. This suggests that keeping a gene-level view of the cell likely enables classifiers to exploit gene–gene dependencies and interaction structure in the latent space, which are blurred or lost under aggressive pooling.

Under cross-experiment transfer, AUROC plateaus around 0.55–0.75, confirming that cross-experiment generalization remains challenging. Within this harder regime, scAraFM representations provide the most robust signal in the low-data region ($\sim$1k–5k training cells), where baseline features degrade more sharply (Supplementary Figure 2), and the CNN+MLP head achieves the best or tied-best AUROC. At full training scale, scAraFM-derived features (PCA_1, ENS with an MLP head) exceed raw baselines; gains are incremental rather than transformative, but most consistent under distribution shift.

### 3.2 ROOT TISSUE

Root datasets exhibit substantial heterogeneity in task difficulty (Supplementary Figure 1, Appendix A.3), with some reaching near-ceiling AUROC under random splits while others remain challenging within-study. Consistent with leaf results, gene-identity-preserving representations (ENS, PCA, CNN) outperform pooled summaries (CLS, WM). Training-size sweeps (Supplementary Figure 3) reveal dataset-dependent scaling, with harder tasks showing steeper gains while easier tasks saturate quickly. Cross-experiment evaluation is not feasible for roots due to lack of matched stress conditions across independent studies; conclusions on this tissue are thus restricted to within-study generalization.

## 4 DISCUSSION

**Cross-study evaluation is the meaningfulness test for transcriptomic embeddings.** The consistent difficulty gradient (random > replicate > cross-experiment) indicates that within-dataset scores may be inflated by study-specific effects (batch, protocol, and label-correlated confounders). We therefore advocate replicate-held-out and cross-experiment protocols as the default yardstick for evaluating transcriptomic representations, aligning evaluation with the intended use of foundation models: transfer to independently generated data. This perspective aligns with the study of Boiarsky et al. (2024), who showed that simple baselines can rival scBERT under within-dataset random splits—a result we interpret as an opportunity for methodological improvement rather than a refutation of representation learning. The key question is whether learned representations degrade gracefully when those shortcuts shift across studies; this is precisely the regime where scAraFM retains the most signal.

**Limitations and future directions.** Our study is limited to *Arabidopsis thaliana* and two tissues (leaf and root), reflecting the current scarcity of large, well-annotated plant scRNA-seq resources compared to human atlases. This constrains both pretraining scale and the diversity of stresses, genotypes, and protocols available for robust cross-study evaluation; in particular, the lack of matched stress conditions across root studies prevents cross-experiment testing in that tissue. Progress will likely require larger and better-curated multi-species corpora, benchmarks that emphasize cross-study generalization over random splits, and representation designs that preserve gene identity while remaining robust across experimental variation.

**Meaningfulness Statement.** A meaningful representation of life must capture transferable biological structure, not dataset-specific artifacts. Stress responses (i.e. how organisms sense, respond to, and adapt under environmental challenges), are fundamental to survival and evolution. Our work demonstrates that meaningful plant transcriptomic representations should preserve gene-level organization and generalize robustly across independently conducted experiments. By prioritizing cross-study evaluation over convenient random splits, we advocate for an empirical standard that rewards models capturing true biological signal rather than protocol shortcuts. Understanding stress adaptation mechanisms through robust computational representations becomes critical for advancing plant biotechnology applications, from targeted crop improvement and precision breeding to engineering stress-resilient cultivars.

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

## A APPENDIX

### A.1 DATASETS

#### A.1.1 DATA PREPROCESSING AND HVG EXTRACTION.

All single-cell RNA-seq datasets were processed with a standardized pipeline implemented in Scanpy to ensure a consistent feature space across samples and to support downstream cross-experiment comparisons. This pipeline was conducted independently for both tissues.

**Gene identifier harmonization and filtering.** Gene identifiers were first harmonized to the Arabidopsis thaliana locus nomenclature. Duplicate gene entries in the expression matrix were collapsed by summing counts across duplicated rows, yielding a one-gene–one-feature representation. We then filtered features to retain only canonical Arabidopsis locus identifiers. Organellar loci and non-target identifiers were excluded from the feature set (including chloroplast-prefixed loci such as ATC, and long non-coding RNA identifiers such as ATHLNC) to avoid expanding the feature space with dataset-specific or non-comparable annotations.

**Library-size normalization and log transformation.** To correct for sequencing-depth differences across cells, raw counts were normalized on a per-cell basis to a fixed library size (counts per 10,000) using Scanpy's total-count normalization, followed by log-transformation with log1p.

**Highly variable gene discovery.** Within each dataset, highly variable genes (HVGs) were identified using Scanpy's dispersion-based procedure with the Seurat flavor, which bins genes by mean expression and ranks genes by normalized dispersion within bins to prioritize biologically informative variation while controlling for mean–variance dependencies.

**Consensus HVG construction across datasets.** To enable cross-experiment analyses in a shared feature space, HVGs were computed independently per dataset and then aggregated into a consensus list using a frequency-based criterion: genes were ranked by the number of datasets in which they were selected as HVGs, and the top $N$ genes were retained (default $N = 4000$). This approach emphasizes genes that are reproducibly variable across studies rather than dataset-specific artifacts. Final verification ensured that the resulting feature set was valid across all processed files (i.e., compatible with a strict intersection in downstream model inputs).

A.1.2 DATASETS USED IN EXPERIMENTAL DESIGN

Supplementary Table 1: **Overview of the datasets employed for downstream stress prediction tasks.**

| Tissue | Dataset ID | # Samples | Stress (# conditions) | # Genotypes | # Replicates |
|--------|-----------|-----------|-----------------------|-------------|--------------|
| Leaf | ERP132245 | 2018 | Drought Response (2) | 1 | 1 |
| Leaf | GSE226826 | 67961 | Pseudomonas syringae Infection (2) | 1 | 1 |
| Leaf | GSE273033 | 50797 | Drought Response (2) | 1 | 2 |
| Leaf | SRP398011 | 22882 | Pseudomonas syringae Infection (2) | 1 | 1 |
| Root | GSE235495 | 24474 | Osmotic Stress (2) | 1 | 2 |
| Root | SRP148288 | 24369 | Induced with estradiol (2) | 3 | 1 |
| Root | SRP166333 | 16949 | Heat Shock (2) | 1 | 1 |
| Root | SRP169576 | 35665 | Sucrose Response (2) | 1 | 1 |
| Root | SRP285817 | 17553 | Temporal Gravitropic Reorientation (3) | 2 | 1 |

A.1.3 DATASETS USED FOR PRETRAINING

Supplementary Table 2: **Collected datasets used for pretraining the leaf and root models.**

| (a) Leaf datasets. | | | (b) Root datasets. | |
|---|---|---|---|---|
| **Dataset ID** | **# Samples** | | **Dataset ID** | **# Samples** |
| CRA002977 | 10,947 | | GSE262840 | 172,776 |
| DRP009643 | 15,400 | | SRP171040 | 33,956 |
| GSE184511 | 23,729 | | SRP173393 | 13,252 |
| GSE226097 | 59,842 | | SRP182008 | 13,514 |
| SCP2703 | 500,348 | | SRP235541 | 27,798 |
| SRP247828 | 93,300 | | SRP267870 | 209,482 |
| SRP253497 | 7,242 | | SRP273996 | 10,431 |
| SRP280069 | 49,016 | | SRP285040 | 1,206 |
| SRP292306 | 5,947 | | SRP330542 | 22,606 |
| SRP307169 | 17,441 | | SRP332285 | 16,017 |
| SRP338044 | 50,418 | | SRP394711 | 356,513 |
| **Total** | **833,630** | | **Total** | **877,551** |

A.2 EVALUATION & FINETUNING PIPELINE

This appendix describes the evaluation workflow used across all experiments and clarifies how train/test splits and caps are constructed for each setup.

**Common pipeline.** For each dataset, we apply the following steps: (i) Preprocessing. We start from preprocessed expression matrices (as described in Section A.1.1), ensuring a consistent feature space and labeling across experiments. (ii) Class balancing. Before defining splits, we balance the data by class by downsampling each class to the size of the least represented class (i.e., balanced-per-class subset). This yields a class-balanced pool used for all subsequent split procedures. (iii) Experimental split definition. We construct train/test partitions according to the split protocol (Random, Replicate, or Cross-Experiment; detailed below). When applicable, the test set is capped to avoid excessively large evaluations and to keep results comparable across datasets. (iv) Training sample capping. From the training partition, we run learning curves by capping the number of training examples to a specified training count (or using ALL available training data). This capping is performed stratified by class to preserve balance. (v) Model training and evaluation. For each representation variant and classifier, we train on the capped training set and report metrics on the fixed test set for that experimental configuration.

**Random split.** We perform a class-balanced random split, with a test size of 20% of the balanced pool. To prevent overly large test sets, we apply the following rule: Let $N$ be the total number of samples after class balancing. Define the target test size as $0.2N$. If $0.2N > 2,000$, we cap the test set to $2,000$ samples, using stratified sampling to preserve class balance. Otherwise, we use the full $0.2N$ balanced test set. Equivalently: $|D_{test}| = \min(0.2N; 2,000)$ with stratified selection when capped.

**Replicate split.** We split by biological/technical replicate to prevent overlap between replicates across train and test. Test set: the first replicate (according to the replicate ordering defined in the dataset metadata). If the test replicate contains more than 2,000 samples, we cap it to 2,000 using stratified sampling by class. Training set: all remaining replicates (from the balanced pool), subsequently subject to the training-count capping described above.

**Cross-experiment split.** We evaluate cross-experiment transfer by training on one dataset and testing on another dataset that shares the same label space. Test set: the entire target dataset (the "other" dataset in the pair), after applying the same preprocessing and class balancing procedure. If the target dataset has more than 2,000 samples, we cap the test set to 2,000 using stratified sampling by class. Training set: the source dataset (balanced), then capped by training count (or ALL).

### A.3 ROOT RESULTS AT MAXIMUM TRAINING COUNT

We investigate the models perfomance on the curation of stressed root datasets in Supplementary Figure 1.

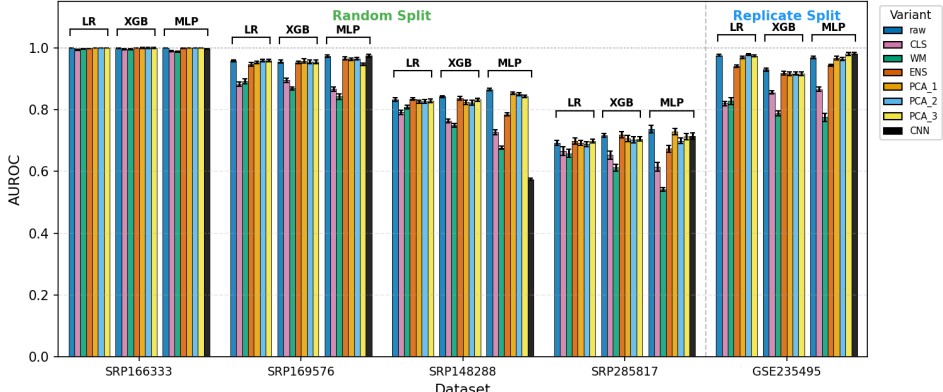

Supplementary Figure 1: **Stress classification performance for *Arabidopsis* root datasets.** Each bar depicts AUROC for each of the classifiers implemented (LR, XGBoost, MLP and CNN+MLP) along raw counts and varying embedding combination methodologies. On the columns, the classification tasks are shown in rising difficulty: random splits and replicate splits. Error bars are shown from bootstrapped test samples.

### A.4 PERFORMANCE IS TIED TO SAMPLE COUNT

We aggregate the results of the representation strategies evaluated when combined with the classifier heads in Supplementary Figure 2 for leaves and Supplementary Figure 3 for roots.

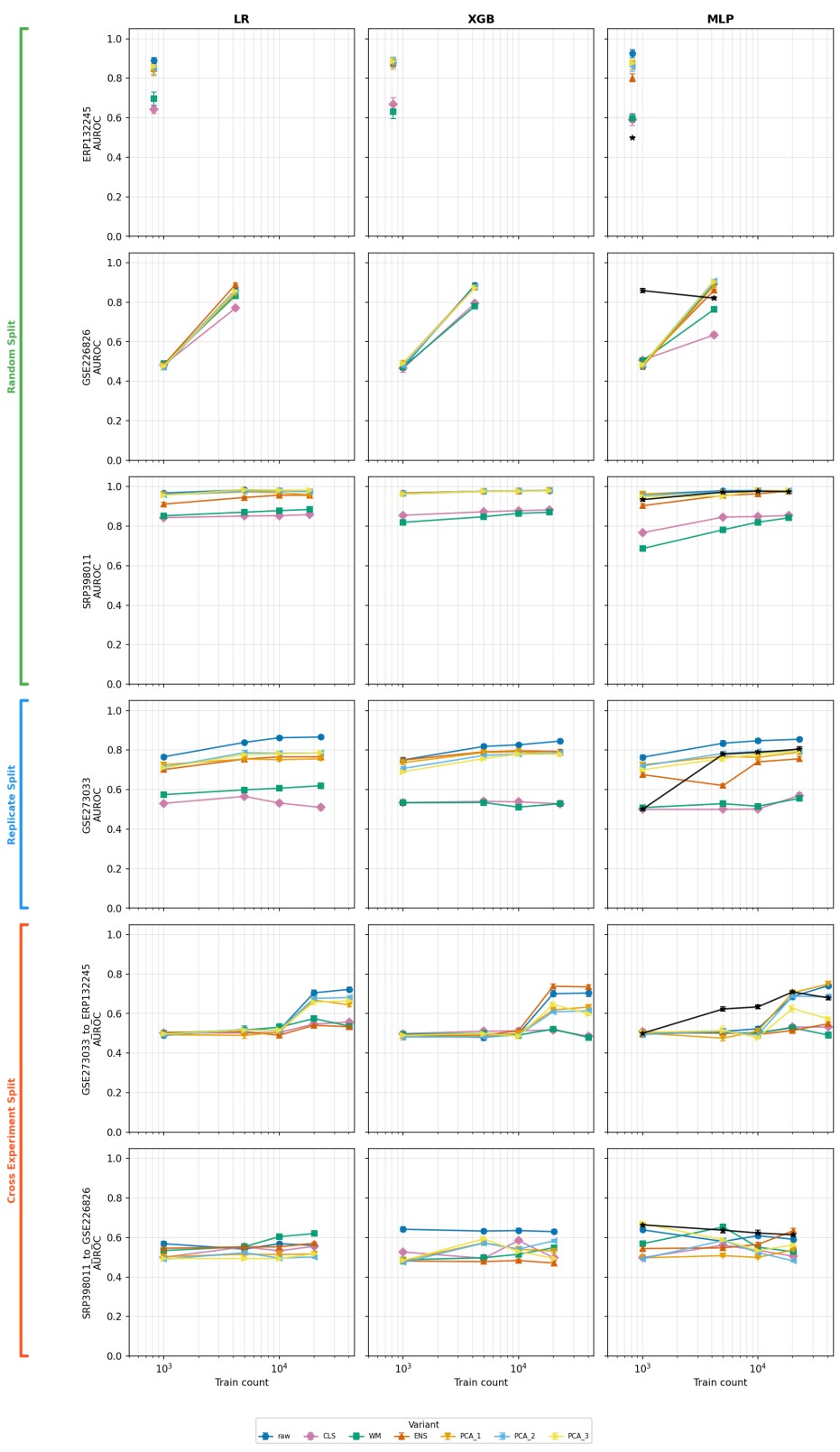

Supplementary Figure 2: **Stress classification performance on varying scenarios for *Arabidopsis* leaves datasets scaling training samples count.** Each column depicts AUROC for each of the classifiers implemented (Logistic Regression, XGBoost, MLP and CNN) along raw counts and varying embedding combination methodologies. On the rows, the classification tasks are shown in rising difficulty: random splits, replicate splits and cross-experiment splits. Error bars are shown from bootstrapped test samples.

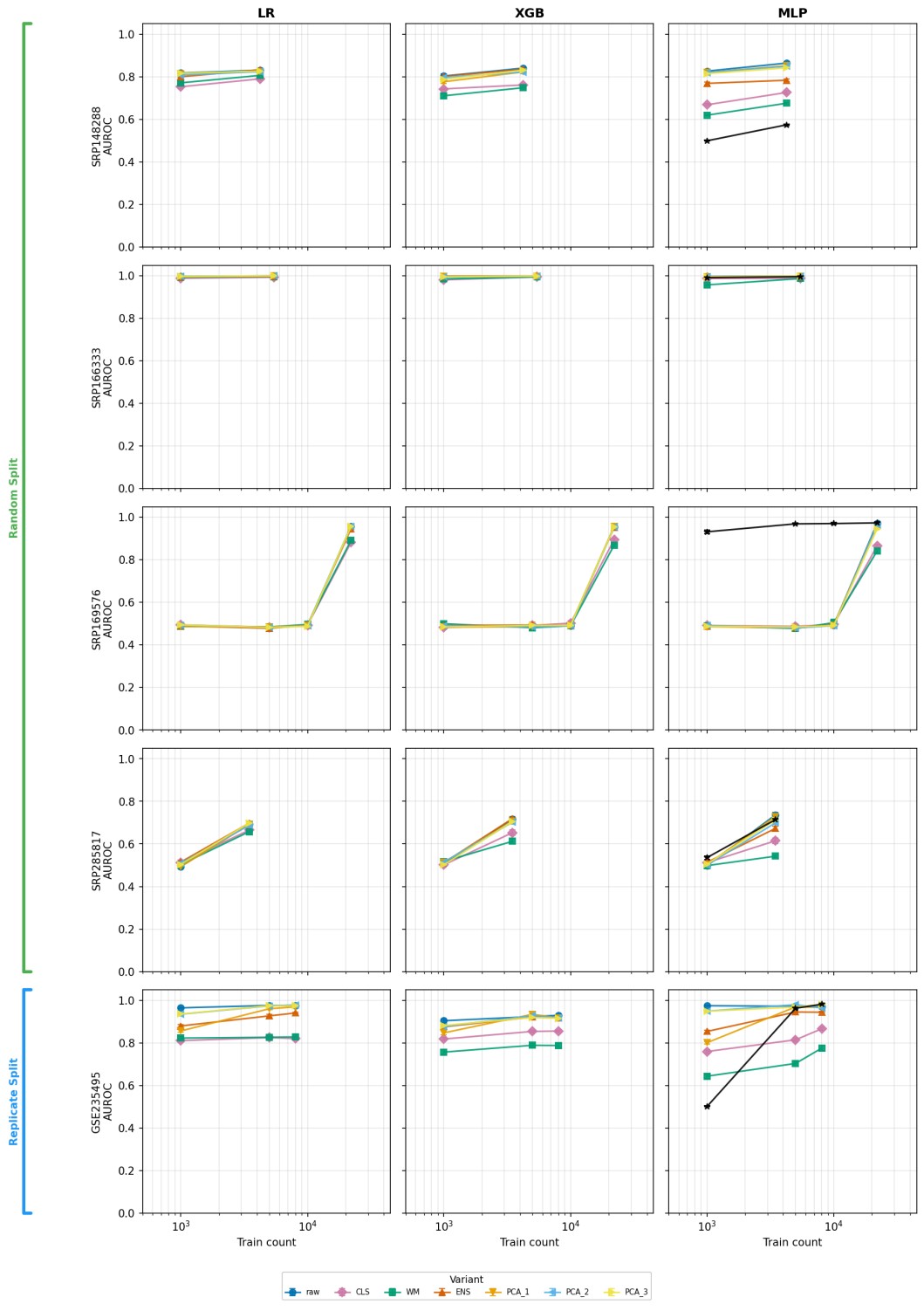

Supplementary Figure 3: **Stress classification performance on varying scenarios for *Arabidopsis* roots datasets scaling training samples count.** Each column depicts AUROC for each of the classifiers implemented (Logistic Regression, XGBoost, MLP and CNN) along raw counts and varying embedding combination methodologies. On the rows, the classification tasks are shown as: random splits and replicate splits. Error bars are shown from bootstrapped test samples.

