# OpenReview forum: "On The Robustness of scRNA-seq Foundation Models for Plants Under Cross-Domain Experimental Shift"
_ICLR.cc/2026/Workshop/LMRL — ICLR 2026 Workshop LMRL Poster_

### Official Review · Reviewer_Ddk4 · 2026-02-23
**Solid Evaluation Framework for Plant scRNA-seq, but Falls Short on Novelty, Benchmarking Tools, and Presentation**

**Rating:** 4
**Confidence:** 4

**Review:**

# On the Robustness of scRNA-seq Foundation Models for Plants Under Cross-Domain Experimental Shift

## Summary
The manuscript introduces scAraFM, an Arabidopsis thaliana-specific foundation model, and evaluates its robustness in stress prediction tasks using scRNA-seq data. The authors benchmark the model against classical machine learning baselines across three increasingly stringent data-splitting protocols: single-experiment random splits, single-experiment replicate splits, and cross-experiment transfer learning. The core findings suggest that random train/test splits artificially inflate model performance by 20 to 30 AUROC points. Furthermore, the study asserts that traditional baselines trained on raw counts frequently rival learned representations , and that retaining per-gene feature structure yields better downstream performance than condensed, pooled summaries

## Strengths
* Rigorous Evaluation Framework: The introduction of a three-tiered splitting strategy (random, replicate, cross-experiment) effectively illustrates the vulnerability of scRNA-seq models to study-specific confounders. This is a necessary and practical framework for the field.
* Honest Baseline Comparisons: The authors provide a candid assessment by thoroughly comparing foundation model embeddings against simple, classical baselines like Logistic Regression and XGBoost on raw counts.
* Actionable Insights on Representations: The empirical finding that gene-identity-preserving features (ENS, PCA, CNN) consistently outperform pooled vectors (CLS, WM) provides a clear directive for downstream feature extraction.

## Weaknesses and Areas for Improvement

**1. Incremental Conceptual and Architectural Novelty**
While the application to plant systems is valuable, the methodological novelty of the work is limited. The model itself, scAraFM, inherits its architecture directly from scBERT without significant algorithmic innovation. Furthermore, the community already has access to plant-specific foundation models, such as scPlantLLM, which the authors themselves cite. Additionally, the dangers of random splits and the profound impact of batch effects are well-documented phenomena in standard scRNA-seq literature, diluting the novelty of the core claims.

**2. Missed Opportunity for a Standardized Benchmark Suite**
The authors successfully diagnose a major issue: the lack of study-aware validation. But stop short of providing a community solution. To elevate the impact of this work, the authors should consider formalizing their curated datasets and evaluation protocols into a structured, downloadable benchmarking suite (akin to GLUE for NLP). Proposing a standardized framework would transition the paper from simply observing a known problem to actively solving it.

**3. Shallow Discussion on Representation Aggregation**
The observation that aggressive pooling (CLS, WM) degrades performance compared to per-gene structure is arguably the most interesting technical takeaway. However, the manuscript only briefly hypothesizes that keeping a gene-level view enables classifiers to exploit gene-gene dependencies. The paper would benefit greatly from a deeper, mechanistic discussion: Why exactly does the CLS token fail to capture this in the transformer's latent space? How can we use this insight to design better, more computationally efficient aggregation layers in future foundation models?

**4. Data Visualization and Readability**
The presentation of the results in the figures (particularly Figure 2, Supplementary Figure 2, and Supplementary Figure 3) is difficult to parse and represents an inefficient use of space. While maintaining a zero-baseline on the y-axis of a bar chart is a correct statistical practice, the current grid layout renders the individual bars too small. Relying purely on color-coding across such a dense matrix of results hinders readability. The authors should strongly consider alternative visualizations, such as line plots with error bands or more condensed summary dot-plots, to make performance scaling and model comparisons more accessible to the reader.

## Conclusion
This paper presents a solid, albeit incremental, evaluation of foundation models in plant transcriptomics. While the core idea of enforcing stricter evaluation protocols is highly relevant, the lack of architectural novelty and the difficulty in reading the figures hold the manuscript back. Addressing the visual presentation and expanding the work into a formalized benchmarking suite would significantly strengthen the submission.

---

### Official Review · Reviewer_F1NM · 2026-02-24
**Good motivation but still has a lot of room for improvement.**

**Rating:** 4
**Confidence:** 3

**Review:**

This study introduces a foundational model of Arabidopsis called scAraFM. The authors evaluated scAraFM in a stress prediction task and found that 1) the baseline of raw gene expression was quite strong, and 2) cross-study evaluation was significantly more difficult compared to randomized splits. I think the comparison between randomization, cross-replication, and cross-experimentation is very important, but the comparisons can be performed a lot better, and there is practically no detail on how scAraFM was trained.

Advantages:

1. Cross-experiment evaluation is a true test of model utility, so it’s really good to see that highlighted.
2. Curated datasets seem to be a useful resource for the community

Weaknesses:

1. Curated data sets is not open-sourced
2. There is little or no pre-training information for scAraFM.
3. Comparisons of randomized, repetitive, and cross-experimental do not control for the dataset. For example, in figure 2, none of the datasets used in the random split was used for replicate split. In this case, one cannot make the claim that replicate split is harder, because it may just be that the dataset used in the replicate split is an inherently harder due to other factors. Similarly, none of the replicate split dataset (specifically GSE273033) was used to make predictions on in the cross experiment split, it was only used for training. This again makes it difficult to say that cross-experiment is harder than cross-replicate, since we never actually see how hard it it to train on another experimental dataset and predict on GSE273033.

---

### Meta-Review · Area_Chair_C2PF · 2026-02-28

**Recommendation:** Accept (Poster)
**Confidence:** 3

**Metareview:**

The reviewers didn't think that this paper met the threshold for acceptance, but given that this is the tiny paper track and both praised the rigorous evaluation & focus on experiment-level splits, I think this is worth discussing at the workshop.

---

### Decision · Program_Chairs · 2026-03-02

**Decision:**

Accept (Poster)

**Comment:**

Please see the meta-review.